# Retinal Proteomic Analysis in a Mouse Model of Endotoxin-Induced Uveitis Using Data-Independent Acquisition-Based Mass Spectrometry

**DOI:** 10.3390/ijms23126464

**Published:** 2022-06-09

**Authors:** Jing Zhang, Jiangmei Wu, Daqian Lu, Chi-Ho To, Thomas Chuen Lam, Bin Lin

**Affiliations:** 1School of Optometry, The Hong Kong Polytechnic University, Kowloon, Hong Kong SAR, China; jing0114.zhang@connect.polyu.hk (J.Z.); jiangmei.wu@polyu.edu.hk (J.W.); da-qian.lu@connect.polyu.hk (D.L.); chi-ho.to@polyu.edu.hk (C.-H.T.); 2Centre for Eye and Vision Research (CEVR), Hong Kong SAR, China; 3Research Centre for SHARP Vision (RCSV), The Hong Kong Polytechnic University, Hong Kong SAR, China

**Keywords:** SWATH-MS, uveitis, lipopolysaccharide, retina, inflammation

## Abstract

Uveitis is a group of sight-threatening ocular inflammatory diseases, potentially leading to permanent vision loss in patients. However, it remains largely unknown how uveitis causes retinal malfunction and vision loss. Endotoxin-induced uveitis (EIU) in rodents is a good animal model to study uveitis and associated acute retinal inflammation. To understand the pathogenic mechanism of uveitis and screen potential targets for treatment, we analyzed the retinal proteomic profile of the EIU mouse model using a data-independent acquisition-based mass spectrometry (SWATH-MS). After systemic LPS administration, we observed activation of microglial cells accompanied with the elevation of pro-inflammatory mediators and visual function declines. In total, we observed 79 upregulated and 90 downregulated differentially expressed proteins (DEPs). Among the DEPs, we found that histone family members (histone H1, H2A, H2B) and blood proteins including haptoglobin (HP), hemopexin (HPX), and fibrinogen gamma chain (FGG) were dramatically increased in EIU groups relative to those in control groups. We identified phototransduction and synaptic vesicle cycle as the top two significant KEGG pathways. Moreover, canonical pathway analysis on DEPs using Ingenuity Pathway Analysis revealed top three most significant enriched pathways related to acute phase response signaling, synaptogenesis signaling, and eif2 signaling. We further confirmed upregulation of several DEPs associated with the acute phase response signaling including HP, HPX, and FGG in LPS-treated retinas by qPCR and Western blot. In summary, this study serves as the first report to detect retinal proteome changes in the EIU model. The study provides several potential candidates for exploring the mechanism and novel therapeutic targets for uveitis and other retinal inflammatory diseases.

## 1. Introduction

Uveitis is a serious ocular inflammatory condition of uvea, which is responsible for approximately 10–15% of blindness cases in the US [1,2]. It is characterized by eye pain, redness, photophobia, and loss of visual acuity. Uveitis can be categorized into anterior uveitis, intermediate uveitis, posterior uveitis, and panuveitis. The posterior uveitis affects the retina and/or the optic nerve, which probably leads to permanent loss of vision. Uveitis can be triggered by infections from bacteria and viruses [3,4], or by some systemic autoimmune diseases such as Behcet’s disease, lupus, Vogt–Koyanagi–Harada (VKH) disease [5,6,7,8]. However, the exact cause or the initiation of uveitis, and how inflammatory responses lead to retinal malfunction and vision loss, remain largely unknown.

Lipopolysaccharide (LPS) is commonly used to establish endotoxin-induced uveitis (EIU), a widely used animal model for studying the pathophysiology of uveitis [9]. LPS is a component of the outer envelope of all Gram-negative bacteria and acts as a strong pro-inflammatory endotoxin. LPS can promote the activation of immune cells by the Toll-like receptor 4 (TLR4)-dependent pathway, leading to the overproduction of multiple pro-inflammatory cytokines, including interleukin 1β (IL-1β), interleukin 6 (IL-6), and inducible nitric oxide synthase (iNOS) [10]. These pro-inflammatory factors can further contribute to the breakdown of the blood–retinal barrier (BRB), and the recruitment and infiltration of immune cells into the retina [9]. The inflammatory response peaks at around 24 h after LPS administration [9,10]. Although the EIU model is used to explore acute ocular inflammation or test the efficiency of drugs for more than 40 years, we still know very little about the mechanism of retinal dysfunction due to LPS-induced inflammation.

Proteomics is an approach to identify and quantify the entire set of proteins in a tissue and serves as a powerful screening tool for exploring the protein expression patterns by mass spectrometry [11,12,13]. Previously, several studies tried to identify protein biomarkers of uveitis from ocular fluids, such as vitreous, aqueous, and tears through proteomic technology [14]. Bahk et al. reported their findings in the vitreous bodies of rats with EIU using two-dimensional gel electrophoresis (2-DE) and microLC-MS/MS and identified crystallin family proteins as major altered proteins in uveitic vitreous [15]. However, there has been no report about retinal protein profile alterations induced by EIU so far. Recently, Sequential Window Acquisition of all Theoretical Mass Spectra (SWATH)-MS has been developed as a novel data-independent acquisition method that enables quantitative, sensitive, and reproducible proteomic analysis [16]. In this study, we investigated the retinal protein profile of C57BL/6J mice after systemic LPS administration by using this novel SWATH-MS approach.

## 2. Results

### 2.1. Retinal Evaluation of the EIU Mouse Model

We induced the EIU model by giving a single intraperitoneal injection of LPS with a dosage of 5mg/kg, which has been widely adopted in studies concerning neuroinflammation [17]. Upon LPS exposure, immune cells including microglia are activated to produce and release pro-inflammatory factors including IL-1β, IL-6, and iNOS via activating TLR4 [18]. Since microglia are essential for the initiation of retinal inflammatory responses in uveitis [19], we first evaluated the EIU model by assessing microglia activation in retinal sections (Figure 1A) and retinal whole mounts (Figure 1B). Microglia cells were immunostained with Iba-1, a classical marker of microglia. Microglia cells in PBS-treated control groups showed a ramified morphology with long processes and small cell soma. In comparison, microglia cells displayed an amoeboid morphology with retracted thick processes and enlarged cell soma 24 h after LPS administration, indicating the activated form of microglia cells in the retina [20]. In addition, we determined the mRNA expression levels of multiple pro-inflammatory mediators in the retina. We observed that the mRNA levels of IL-1β (*p* < 0.05), IL-6 (*p* < 0.001), iNOS (*p* < 0.01), and TLR4 (*p* < 0.001) were all significantly increased 24 h after systemic LPS administration (Figure 1C). Similarly, we observed that protein expression levels of IL-1β, IL-6, iNOS, and TLR4 were elevated by ELISA (Figure 1D) and Western blot (Figure 1E), Appendix A A-C show three complete Western blot full images. These results indicate a successful induction of inflammatory responses in the retina by systemic LPS administration. In addition, we performed immunohistochemistry and electroretinogram (ERG) to evaluate retinal structure and function changes in the C57BL/6J mouse model of uveitis. After systemic LPS administration, we did not see an obvious difference in terms of nuclear layer thickness on H&E-stained retinal sections between LPS-treated groups and PBS-treated controls (Appendix A). However, we observed a dramatic reduction in electrical activities in response to flashlights under either dark- or light-adapted condition, including a decrease in amplitudes of a-wave (scotopic 0.1, 1, 3 cd·s/m^2^ and photopic 10 cd·s/m^2^) and B-wave (scotopic 0.01, 0.1, 1, 3 cd·s/m^2^ and photopic 3, 10 cd·s/m^2^) (Figure 1F,G). Collectively, these results demonstrate that systemic LPS administration can induce an inflammatory response and a severe loss of visual function in the retina.

### 2.2. Ion Library Generation Using a Pooled Retinal Proteome

Approximately 1.5μg retinal peptides from all PBS- (n = 4) or LPS-treated (n = 4) mice were pooled together to generate an ion library for SWATH analysis. A total of 3397 non-redundant proteins at global 1% false discovery rate (FDR) and 29,066 distinct peptides were found by ProteinPilot 5.0.1 (Sciex) (Appendix A for the protein and peptide FDR analysis, Appendix A for the list of proteins identified, respectively). This ion library serves as the database to support quantitative analysis in the following SWATH acquisition.

### 2.3. Quantitative Analysis of Differentially Expressed Proteins

Differentially expressed proteins (DEPs) between PBS- and LPS-treated mouse retinas were quantified and compared based on the same ion library after retention time alignment and normalization. In summary, a total of 79 proteins were significantly upregulated, and 90 proteins were significantly downregulated in LPS- versus PBS-treated mouse retinas (fold change (FC) > 1.4 or <−1.4, *p* < 0.05, unpaired *t*-test). The volcano plot showed all of these DEPs between the two groups (Figure 2). We summarized all DEPs in Appendix A and listed significance- and FC-based top 10 upregulated DEPs and downregulated DEPs in Table 1. Among the DEPs, the proteins of histone family members including histone H1, H2A, and H2B were increased after LPS stimulation and were listed as the top upregulated proteins (Figure 2 and Table 1). In addition, several blood proteins such as haptoglobin (HP), hemopexin (HPX), fibrinogen gamma chain (FGG), and fibrinogen alpha chain (FGA) were also dramatically elevated in the retina induced by EIU (Figure 2 and Table 1).

We further conducted functional analysis for all DEPs based on PANTHER gene classification analysis (http://www.pantherdb.org/ (accessed on 2 November 2021)). According to gene ontology (GO), the top three molecular functions were “binding” (48.9%), “catalytic activity” (30.5%), and “molecular function regulator” (5.7%) (Figure 3A). In the biological process, the top three included “cellular process” (32.6%), “metabolic process” (19.7%), and “biological regulation” (16.8%) (Figure 3B). The cellular components distribution was “cellular anatomical entity” (43.3%), “intracellular” (37.1%), and “protein-containing complex” (19.6%) (Figure 3C).

### 2.4. Pathway Analysis of Differentially Expressed Proteins

We uploaded all DEPs to the STRING database (https://string-db.org/, accessed on 25 November 2021) to identify the crosstalk among them. In total, there were 167 nodes (gene name) and 345 edges (predicted functional associations) observed in the network (Figure 4). Based on the KEGG database, the top two enriched pathways included “phototransduction” (mmu04744, 4 red nodes in Figure 4, strength: 1.32, FDR: 0.006) and “synaptic vesicle cycle” (mmu04721, 8 blue nodes in Figure 4, strength: 1.17, FDR: 0.000). Other enrichment KEGG pathways are presented in Appendix A.

To identify the interactions among these DEPs and pathways involved, we conducted “canonical pathway analysis” using Ingenuity Pathway Analysis (IPA) software. The canonical pathways that were identified are listed in Appendix A, and the top pathways are listed in Figure 5A, which included “synaptogenesis signaling pathway”, “acute phase response signaling”, “eukaryotic initiation factor 2 (eif2) signaling”, etc. The fold change of proteins involved in these pathways is illustrated in heatmaps in Figure 5B. We also found many molecules in these pathways clustered together in the STRING network map, suggesting their close relationships, and we highlighted them in circles with different colors (Figure 4). In addition, we found that 7 out of 10 proteins involved in “retinal degeneration” showed significant downregulation during the disease and function analysis of IPA (Figure 5B).

### 2.5. Validation of Proteins in Acute Phase Response Signaling

LPS is a potent inducer of acute phase proteins, which are blood proteins and are also closely related to inflammatory responses. Interestingly, we found that the “acute phase response signaling” is the only signaling that showed a prediction z-score higher than 2.0 in IPA. Thus, we next selected three proteins (HP, HPX, and FGG) in this pathway for validation by qPCR and Western blot analysis using three independent retina samples. Only the mRNA expression level of HP (*p* < 0.01) was elevated in the retina from LPS groups versus PBS groups, but no significant difference was identified in mRNA expression levels of HPX and FGG between the two groups (Figure 6A). Interestingly, Western blot results showed that the protein levels of HP, HPX, and FGG were all significantly upregulated in LPS groups compared to those in control groups (Figure 6B; original blot images in Appendix A–G), which were consistent with our SWATH-MS findings.

## 3. Discussion

Endotoxin-induced uveitis (EIU) is a good model to study inflammatory responses due to the breakdown of the blood–retinal barrier induced by LPS and pro-inflammatory cytokines [9]. EIU-induced inflammation can lead to the infiltration of immune cells from blood circulation, the activation of microglial cells, and the production and release of pro-inflammatory factors in the retina [9]. However, how EIU-induced inflammation leads to microglia activation and function declines in the retina remains largely unknown. In this study, we investigated the retinal proteome of acute retinal inflammation induced by LPS and identified several potential proteins and signal pathways that may be crucial for retinal dysfunction caused by EIU.

Using a highly sensitive SWATH-MS-based proteomics profiling approach, we identified several histone family members, including histones H1, H2A, and H2B, as the top upregulated proteins in the retina after LPS stimulation. Histones are well known for their ability to regulate DNA condensation and gene transcription. However, histones are also present in the cytoplasm and extracellular space during host defense and inflammatory injury [21]. Both H2A and H2B can neutralize endotoxin and show antimicrobial activity against E. coli in the human placenta [22]. Augusto et al. have found that histones H2A and H4 are able to bind with LPS, which can reduce the production of inflammatory factors TNF-alpha and nitric oxide in macrophages [23]. On the other hand, other previous studies also have shown that histones can act as an inducer of inflammation and cause damage or even the death of host cells during lung injuries [24]. In the retina of LPS-induced uveitis, we postulate that the significant elevation of histones might serve as a defender to neutralize LPS [21]. However, whether the overexpression of histones (over 10 times in protein level) would cause cytotoxicity to retinal cells remains to be investigated.

Through the IPA analysis, we identified “acute phase response signaling” as the top canonical pathway with a z-score of 2.111. Additionally, molecules in this pathway including HP, HPX, FGG, and FGA were also ranked top in those upregulated DEPs. HPX and HP are synthesized in the liver and released in high concentration to combine with free heme and free hemoglobin to protect the body from iron- and heme-mediated oxidative damage during hemolysis. The blood–retinal barrier (BRB) can prevent HPX and HP from entering the retina [25]. Overexpression of HPX has been reported in the vitreous fluid of diabetic macular edema patients by proteomic analysis, probably due to leakage from the BRB [26]. However, photoreceptors and ganglion cells also have the potential to express HP and HPX locally in the retina [25]. In the intracerebral hemorrhage (ICH) model, the deletion of HPX aggravates brain injury and intraperitoneal treatment of hemopexin can reduce the early-stage blood–brain barrier (BBB) disruption and cell death [27,28]. In addition, the genetic deletion of HP in young mice can reduce brain damage, improve neurological function, but show aggravated outcomes in aged HP KO mice after ICH [29]. Thus, HPX and HP are closely related to the BBB and BRB, and possibly play a protective role in the central nervous system during hemorrhage. Fibrinogen is the circulating precursor of fibrin, which is involved in wound healing, inflammation, and fibrinolysis. Schachtrup et al. have found that fibrinogen deposits in the spinal cord after injury and inhibits the neurite outgrowth [30]. A series of studies concerning Alzheimer’s disease (AD) reported a correlation of AD with fibrinogen deposition [31,32]. Merlini et al. have shown that fibrinogen promotes spine elimination and cognitive deficits by microglia activation through CD11b-CD8, and genetic deletion of fibrinogen binding motif to CD11b can decrease neuroinflammation, synaptic deficits, and cognitive decline in AD model mice [31]. In our study, we confirmed the upregulation of HP, HPX, and FGG using Western blot analysis, but only HP mRNA was found to be elevated 24 h after LPS stimulation. Yet, the reason for the discrepancies is unclear. The mRNA expression does not necessarily infer the exact abundance of proteins indicated [33]. One possible reason for the discrepancies is that the HPX and FGG mRNA alteration reaches a peak in less than 24 h after systemic LPS stimulation and then falls to baseline level after 24 h, the time point that we collected retina tissues. Another possibility is that the BRB, which separates retinal tissue from circulating blood, could be potentially disturbed in response to LPS stimulation [26], causing the entry of plasma proteins, such as HPX and FGG, from blood into the retina through the BRB and leading to elevation of the plasma proteins in the retina of LPS-induced uveitis. It warrants further investigation to confirm either of the possibilities in the future.

The synaptic function is crucial for proper visual signal transduction. The outer plexiform layer (OPL) and inner plexiform layer (IPL) are two major synaptic layers of the retina. Synaptogenesis and synaptic plasticity are important during retinal development and aging processes. Tian et al. reported that the synapse formation of the retina initiates before the eye open and reaches a peak at P21 in mice [34]. During aging processes, the impairment of synaptic plasticity could be observed, which is accompanied with a progressive increase in inflammation [35]. In the central nervous system, neuroinflammation can be detrimental to neurons by affecting synaptic proteins. Neurodegenerative diseases like AD are associated with increased neuroinflammatory markers (e.g., IL-1β and TNF-α) and loss of synaptic markers (e.g., synaptophysin and drebrin) [36]. In addition, Witcher et al. found that microglia depletion in the brain can prevent the transcriptome alteration in the synaptogenesis signaling induced by the traumatic brain injury (TBI) model, implicating a relationship between inflammation and synaptogenesis via microglia [37]. Based on our bioinformatic analysis, we found that “synaptic vesicle cycle” in KEGG analysis and “synaptogenesis signaling” in IPA analysis ranked top in the altered pathways in the EIU model. Thus, the synaptic dysfunction probably accounts for the visual impairment in the EIU model, leading to marked reduction in the a-wave and B-wave amplitudes of ERG.

The eif2 signaling pathway was also identified as the top-upregulated canonical pathway in our study. Mammalian cells can induce the eif2 signaling following infection by bacterial pathogens, and the transcription of cytokines including TNFα is regulated by eif2 signaling [38]. Khan et al. also observed an alteration in eif2 signaling upon LPS administration in healthy human leukocytes [39]. Some studies reported a close relation of eif2 signaling with neuron function including neuronal degeneration in AD [40]. Cagnetta et al. recently found that eif2 signaling affected the axon growth and neural wiring of retinal ganglion cells, suggesting that the eif2 pathway could be a therapeutic target for neural repair [40]. Consistently, retinal cells may also respond to LPS stimulation with an upregulation of eif2 signaling.

We also identified that a series of genes related to retinal phototransduction and retinal degeneration were downregulated. The phototransduction cascade is mediated by a single G-protein transducin (Gt) consisting of three subunits Gtα (encoded by gene Gnat1), Gtβ, and Gtγ (encoded by gene Gngt1). The Guanylate cyclase-activating protein 1 (GCAP1, encoded by Guca1a) is involved in the calcium-dependent regulation of guanylate cyclases in rods and cones to adjust flash sensitivity [41,42]. The mutation of GUCA1A has been identified to be associated with cone dystrophies and maculopathy [41,43,44]. Therefore, the decreased expression of these phototransduction-related proteins including Gtα, Gtγ, and GCAP1 in our EIU model may contribute to retinal function declines through attenuation of the phototransduction process.

Crystallin is a major protein of the lens and can be further categorized into α, β, and γ subtypes. It is also expressed in the retina [45,46]. Crystallin proteins are reported to be dramatically elevated in experimental autoimmune uveitis (EAU)-induced retinal mitochondrial protein, EAU-induced aqueous and vitreous protein, and EIU-induced vitreous protein profiles [15,47,48], and are suggested as potential biomarkers and therapeutic targets for uveitis [14]. In our study, we also observed an elevation of crystallin proteins including crystallin alpha A (CRYAA, FC = 2.627, *p* = 0.183), crystallin gamma E (CRYGE, FC = 2.050, *p* = 0.324), crystallin beta B3 (CRYBB3, FC = 1.972, *p* = 0.053), crystallin beta A1 (CRYBA1, FC = 1.773, *p* = 0.239), etc. Even though most of these proteins showed fold changes larger than 1.4, they were not identified as DEPs due to *p*-values higher than 0.05. A larger sample size in the future study may provide us with more reliable data to confirm whether crystallin families are also involved in this process.

To the best of our knowledge, this is the first study to explore the retinal proteome of EIU in animals. Additionally, we also need to point out several limitations of this study. First, the EIU protocols are suitable for various routes including intraocular, intraperitoneal, tail vein, footpad, and subcutaneous injections of LPS [9]. Here, we are more interested in how the retina responds to systemic inflammation. Moreover, systemic administration of LPS can offer an opportunity to investigate alterations in the BRB and microenvironment changes during neuroinflammation. Thus, we chose the intraperitoneal route for EIU induction and used a dosage of 5mg/kg, a similar dosage to that used in other studies [49,50,51]. We observed a robust change in protein profile at such dosage. Various dosages and multiple time-points are probably needed to have a clearer picture of the whole regulation process in the retina by EIU. Second, our current study only conducted verification for proteins involved in acute phase response signaling, since it is the only canonical pathway with a z-score larger than 2.0. During the data mining procedure, we found other interesting signaling pathways altered in the EIU group including the eif2 signaling, synaptogenesis signaling, etc. However, we know very little about the potential role of these proteins in retinal inflammation or uveitis, which warrants further investigation on these pathways in the retina in future work.

## 4. Material and Methods

### 4.1. Animals and LPS Administration

C57BL/6J mice (Stock number: 000664) were obtained from Jackson Laboratory for this study. Mice at 6–8 weeks of age were used. Mice received a single intraperitoneal injection of 5.0 mg/kg body weight lipopolysaccharide (LPS, Sigma-Aldrich, St. Louis, MO, USA) diluted in phosphate-buffered saline (PBS) in LPS-treated groups, and the same volume of PBS in control groups. The mice were sacrificed for retina collection 24 h after LPS administration. All animals were maintained at the Centralized Animal Facilities (CAF) of the Hong Kong Polytechnic University with water and food ad libitum under a 12 h of light and dark cycle. All experimental procedures were approved by the Animal Subjects Ethics Sub-committee (ASESC) of The Hong Kong Polytechnic University and conducted in accordance with the ARVO statement for the use of animals.

### 4.2. Electroretinogram (ERG) Analysis

ERG was recorded using the Celeris ERG system (Diagnosys LLC, Lowell, MA, USA) as described previously by us [20]. After dark adaption for 12 h, the mice were anesthetized with a mixture of ketamine (100 mg/kg) and xylazine (20 mg/kg). The pupils were dilated with 1% mydriacyl (Alcon, Fort Worth, TX, USA) for 5 min before the test, and 3% Hypromellose lubricating gel solution (Alcon, Fort Worth, TX, USA) was applied to the cornea. The scotopic ERG was adopted with a gradual increase in light intensities from 0.01, 0.1, 1, to 3 cd·s/m^2^. After 10 min of light adaption with background light intensity of 30 cd·s/m^2^, a photopic ERG was performed at 3 and 10 cd·s/m^2^ light intensity. The amplitudes of a-wave and B-wave were analyzed by the Celeris ERG software (Diagnosys LLC, Lowell, MA, USA).

### 4.3. Retina Tissue Extraction

After anesthesia, the mice were sacrificed with cervical dislocation. The right eyes of each mouse were enucleated and kept in the ice-cold PBS. Retina tissues were isolated from the cornea, lens, the retinal pigment epithelium (RPE), and choroidal tissues by using sterile scissors and forceps. After being rinsed in PBS, the retina was flash-frozen in liquid nitrogen immediately in a 1.7 mL EP tube and stored at −80 °C for further usage.

### 4.4. Immunostaining

The immunostaining and confocal imaging were conducted as described in our previous study [52]. Briefly, the retina tissue was fixed with 4% paraformaldehyde (PFA) for 1 h. Some retinas were sectioned at a thickness of 15 μm using a cryostat. Both cross sections and whole-mount retinal tissues were incubated in blocking solution (3% donkey serum, 2% bovine serum albumin, and 0.5% Triton-X in PBS) for 1 h at room temperature (RT), and then incubated in solution with primary antibody rabbit anti-Iba1 (019-10741, Wako, 1:500) overnight at 4 °C. The sections or tissues were rinsed in PBS and then incubated in secondary antibody donkey anti-rabbit Alexa Fluor 488 IgG (1:500, A21206, Invitrogen, Waltham, MA, USA) for 1 h at RT. Nuclei were counterstained with DAPI (4′,6-diamidino-2-phenylindole) (Vector Laboratories, Burlingame, CA, USA). After being rinsed in PBS, the sections or whole-mount tissues were mounted with Vectashield mount medium (Vector Laboratories, Burlingame, CA, USA). Some retinal sections were stained with hematoxylin and eosin (H&E). Confocal images were acquired with a Zeiss LSM 800 Upright Confocal Microscope (Zeiss, Oberkochen, Germany) with a pixel resolution of 1024 × 1024, Plan-Apochromat 20 ×/0.8 objectives, 0.5 μm Z-stack.

### 4.5. qPCR

Total RNA was extracted from retina tissues using TranZol Up Plus RNA Kit (TransGen Biotech, Beijing, China) following the manufacturer’s instructions. cDNA was synthesized with High-Capacity cDNA Reverse Transcription Kit (Thermo fisher, Waltham, MA, USA). Then, qPCR was carried out on the QuantStudio 7 Flex Real-Time PCR System (Applied Biosystems, Waltham, MA, USA) using TB Green Premix Ex Taq (TAKARA Bio, Kusatsu, Shiga, Japan). The relative mRNA levels were determined by 2^−ΔΔCt^ method with housekeeping gene GAPDH. The primer sequences of qPCR are listed in Appendix A.

### 4.6. Protein Extraction from Retina Tissues

The retina was transferred to a homogenization tube with prefilled beads (CKMix-2 mL, P000918-LYSK0-A, Bertin, Zone industrielle, Thiron-Gardais, France), and 200 μL of lysis buffer containing 5% sodium dodecyl sulfate (SDS) and 50 mM triethylammonium bicarbonate (TEAB) was added to the tube. The retina sample was homogenized at 4 °C, 5800 rpm for 30 s × 4 cycles with 20 s of intervals in the tissue homogenizer (Precellys Evolution, Bertin, Zone industrielle, Thiron-Gardais, France). The sample was first centrifuged at 15,000 rpm for 5 min at 4 °C, then pipetted the supernatant into a new tube and further centrifuged at 15,000 rpm for 30 min at 4 °C. The supernatant was collected into a new tube and the protein concentration was determined by Pierce Rapid Gold BCA Protein Assay (A53225, Thermo, Waltham, MA, USA) following the manufacturer’s protocol.

### 4.7. S-Trap-Based Peptide Preparation

Aliquots of 50µg of retina protein lysate were used for peptide preparation using the S-Trap method in accordance with previous protocol [53]. Briefly, the lysate was reduced in 20 mM Dithiothreitol (DTT) at 95 °C for 10 min, and then alkylated in 40 mM iodoacetamide (IAA) in the dark for 10 min at RT. Next, the solution was acidified with 12% aqueous phosphoric acid and diluted with S-Trap protein binding buffer. The S-Trap micro spin column (ProtiFi, Farmingdale, NY, USA) was used to trap the protein in the membrane, and protein was digested with trypsin (1:25, *w*/*w*, trypsin: protein) on the filter at 47 °C for 1 h. After digestion, the peptides were eluted with consecutive reagents: (1) 50 mM TEAB, (2) 0.2% aqueous formic acid (FA), and (3) 50% Acetonitrile (ACN) containing 0.2% FA. The eluted solution was dried out with a vacuum concentrator (Labconco, Kansas City, MO, USA) at 4 °C. The peptides were resuspended in 0.1% FA and measured with Pierce Quantitative Colorimetric Peptide Assay (Cat no.23275, Thermo, Waltham, MA, USA) following the manufacturer’s instruction. The peptide sample was normalized with 0.1% FA to a final concentration of 0.5 µg/μL for mass spectrometry identification and quantification.

### 4.8. Protein Identification and SWATH-MS

The MS analysis was conducted on a TripleTOF 6600 system (Sciex, Framingham, MA, USA) with Analyst TF1.7 software (Sciex, Framingham, MA, USA). For ion library building based on information-dependent acquisition (IDA), 1.5 µg of peptide solutions from each retina sample were pooled together, and a total of 2 ug of the peptide mixture was injected to generate the ion library with 2 technical replicates. For SWATH-MS acquisition, 2 µg of each individual peptide sample was injected with 2 technical replicates. Peptides were first loaded onto a trap column (100 μm i.d. × 2 cm, packed with Acclaim PepMap100 C18, 5 μm, 100 Å) with loading buffer (0.1% FA, 2 % ACN) at 2 µL/min for 15 min. Later, the peptides were further separated on a nano-LC column (NanoLC 415, Eksigent). For IDA, the MS spectra were set as 350–1800 *m*/*z* with 250 ms accumulation time. The ions with a threshold over 125 cycle/s will be counted for MS/MS. For SWATH-MS, the TOF-MS was set for a variable isolation window with the mass range from 100 to 1800 *m*/*z* of 100 overlapping variable windows. The accumulation time was 30 ms for each fragment ion, and the total duty cycle was 3.0 s.

### 4.9. Bioinformatic Analysis

Protein identification was conducted using the ProteinPilot 5.0.1 software (Sciex, Framingham, MA, USA). The IDA data were processed by the Paragon algorithms and searched against the mus musculus (mouse) Uniprot database (UP000000589, updated on 4 June 2021, reviewed). The searching parameter settings include trypsin as the enzyme for digestion, IAA for cysteine alkylation, thorough search effort and biological modification, and a 1% FDR as the filter.

We set the pooled protein IDA result as the ion library for SWATH quantification. PeakView 2.2 (Sciex, Framingham, MA, USA) was used to generate and calibrate the peaks with parameter settings as: 10 peptides per protein, 6 transitions per peptide, 90% peptide confidence threshold, 1% FDR, 10 min XIC extraction window, and 75 ppm width. Protein differential expression analysis was carried out with the help of MarkerView 1.3 software (Sciex, Framingham, MA, USA). Additionally, the selection criteria of DEPs were set as: fold change >1.4 or <−1.4, *p*-values < 0.05 (unpaired *t*-test) with at least 2 matching peptides per protein, represented as the mean of fold change of all biological samples. The volcano plot was generated by GraphPad Prism 9.0 (GraphPad Software, San Diego, CA, USA).

For the bioinformatics analysis, the protein and gene IDs were obtained using Retrieve/ID mapping tool on the Uniprot (https://www.uniprot.org/, accessed on 27 October 2021). Functional analysis of gene ontology (GO) annotations of all DEPs was conducted based on PANTHER gene classification analysis (http://www.pantherdb.org/, version 16.0, accessed on 1 December 2020) [54]. The interactions of DEPs were preceded using the search tool for the retrieval of interacting genes and proteins (STRING) database (https://string-db.org/, version 11.5, accessed on 12 August 2021) [55]. The following settings were used for enriched KEGG analysis in the STRING database. Organism: *Mus musculus*; network type: full STRING network; required score: medium confidence (0.400); FDR stringency: medium (5 percent). The statistical background of enrichment analysis was assumed as whole genome.

Ingenuity pathway analysis (Qiagen, Germantown, MD, USA) was performed on all DEPs [56]. A brief schematic workflow for the quantitative proteomics is shown in Figure 7. The mass spectrometry proteomics data have been deposited to the ProteomeXchange Consortium via the PRIDE [57] partner repository with the dataset identifier PXD033532.

### 4.10. Western Blot

The protein sample from the retina was denatured by boiling in SDS sample buffer, and an aliquot of 10ug of total protein was subjected to 10% SDS-PAGE gels, followed by transferring to a PVDF membrane. The membrane was first blocked with 5% non-fat milk in TBST buffer for one hour and then incubated overnight at 4 °C with primary antibodies. After washing in TBST for three times, the membrane was incubated in a blocking solution with secondary antibodies at RT for 1 h. The membranes were washed 5 times and incubated with SuperSignalTM West Pico PLUS Chemiluminescent Substrate (Invitrogen, Waltham, MA, USA) following the manufacturer’s protocol and were ready for detection. The antibodies and concentration used are listed below: rabbit anti-iNOS (ab15323, Abcam, 1:250), rabbit anti-TLR4 (A5258, Abclonal, 1:1000), rabbit anti-HP (ab256454, Abcam, 1:1000), rabbit anti-HPX (A5603, Abclonal, 1:1000), rabbit anti-FGG (A5642, Abclonal, 1:1000), mouse anti-GAPDH (CB1001, Millipore, 1:1000), HRP goat anti-rabbit IgG (AS014, Abclonal, 1:5000) and HRP goat anti-mouse IgG (AS003, Abclonal, 1:5000).

### 4.11. ELISA

Mouse IL-1β ELISA kit (CSB E08054m, CUSABIO, Wuhan, China) and mouse IL-6 ELISA kit (CSB E04639m, CUSABIO, Wuhan, China) were used to quantify IL-1β and IL-6 protein levels, respectively, following the manufacturer’s instructions. In brief, 100 μL of standards and aliquots of protein lysate (1 mg/mL) were added to the corresponding wells and incubated for 2 h at 37 °C, followed by incubating with biotin-antibody for 1 h at 37 °C. After washing the plate for three times, each well was incubated with 100 μL of HRP-avidin for 1 h at 37 °C. After washing the plate three times, 90 μL of TMB substrate was added to each well and incubated for 15–30 min at 37 °C under dark protection. Stop solution was added to each well, followed by reading the plate at 450 nm on a microplate reader (Thermo Scientific Varioskan LUX Multimode Microplate Reader).

### 4.12. Statistical Analysis

The data were expressed as mean ± standard deviation (SD). The unpaired t-test was performed using Microsoft Excel 2021. Statistical comparisons between groups were performed using two-tailed unpaired *t*-test with a *p*-value less than 0.05.

## 5. Conclusions

In summary, we identified dramatical elevation of histone families and several blood proteins in C57BL/6J mice after endotoxin induction by SWATH-MS. These affected proteins may be novel therapeutic candidates for treatment of uveitis and other retinal inflammatory diseases.

## Figures and Tables

**Figure 1 ijms-23-06464-f001:**
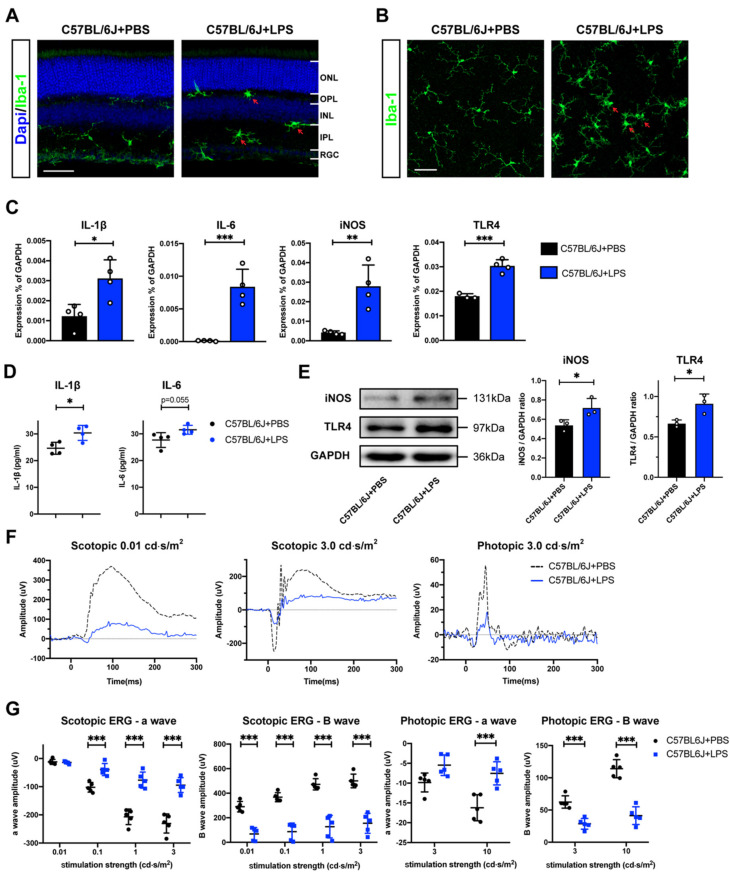
Retinal evaluation of the endotoxin-induced uveitis (EIU) model. Confocal images of retinal sections (**A**) and retina whole-mount focusing on the inner plexiform layer (IPL) (**B**) from C57BL/6J mice 24 h after receiving an intraperitoneal injection of LPS (5 mg/kg) or PBS (blue: Dapi; green: Iba-1; red arrows: activated microglial cells; scale bars, 50 μm). (**C**) Expression levels of mRNA for interleukin 1β (IL-1β), interleukin 6 (IL-6), inducible nitric oxide synthase (iNOS), and toll-like receptor 4 (TLR4) in the retina tissue from PBS and LPS groups (n = 4), (**D**) ELISA analysis of IL-1β and IL-6 expression levels in the retina tissues of LPS- and PBS-treated groups (n = 4). (**E**) Western blotting analysis and densitometry of iNOS and TLR4 in the retina of LPS- and PBS-treated groups (n = 3). GAPDH levels were used as a loading control. (**F**) Representative ERG images of scotopic 0.01 cd·s/m^2^, scotopic 3.0 cd·s/m^2^, photopic 3.0 cd·s/m^2^ responses from C57BL/6J + PBS (black dot line) and C57BL/6J + LPS (blue solid line). (**G**) Average amplitudes of a-wave and B-wave elicited at scotopic 0.01, 0.1, 1, 3 cd·s/m^2^ and photopic 3, 10 cd·s/m^2^ in C57BL/6J + PBS (n = 5) and C57BL/6J + LPS (n = 5). Data are expressed as the mean ± SD. * *p* < 0.05, ** *p* < 0.01, *** *p* < 0.001.

**Figure 2 ijms-23-06464-f002:**
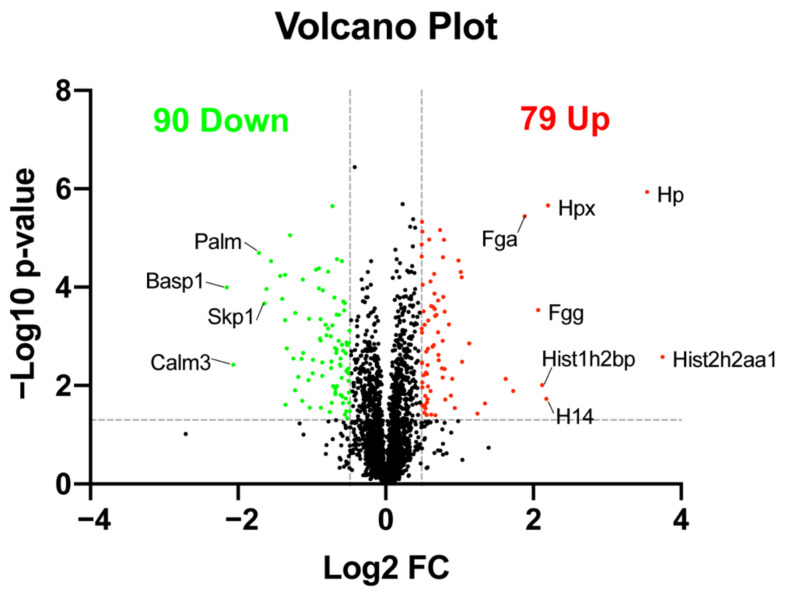
A volcano plot created in GraphPad Prism showing the comparison of DEPs between the treated and control groups. The differential expression was determined as fold-change ratio (LPS/PBS) > 1.4 or <−1.4, *p* < 0.05, unpaired *t*-test. Each point represents a protein. Red dots represent upregulated proteins and green dots represent downregulated proteins.

**Figure 3 ijms-23-06464-f003:**
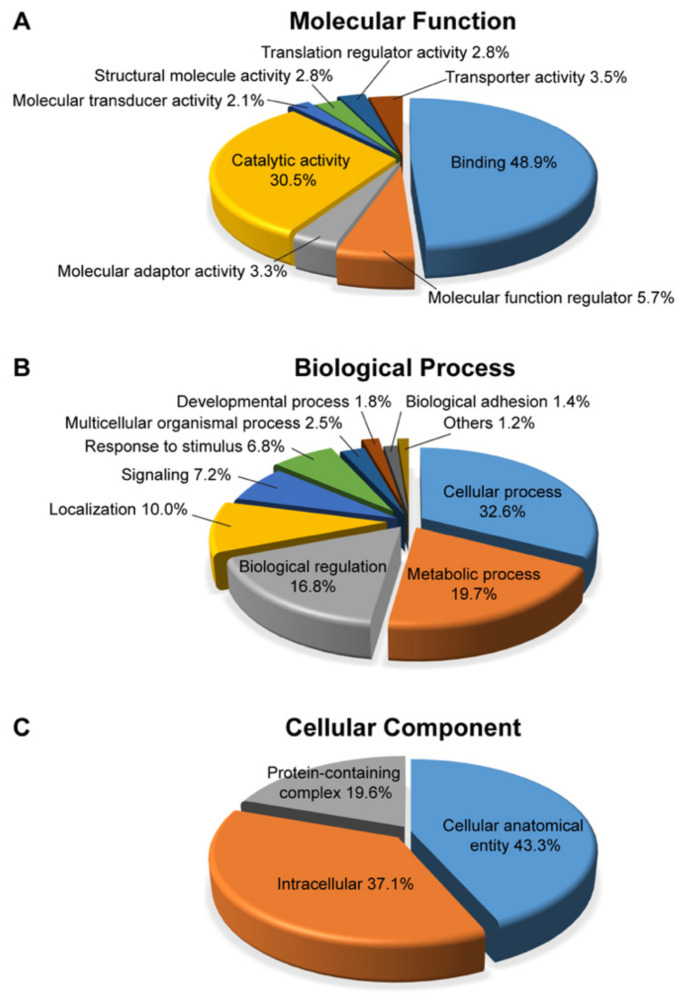
Pie charts showing the proteome distribution. All DEPs were annotated with the PANTHER Classification system (http://www.pantherdb.org/, accessed on 2 November 2021) according to their molecular function (**A**), biological process (**B**), and cellular component (**C**).

**Figure 4 ijms-23-06464-f004:**
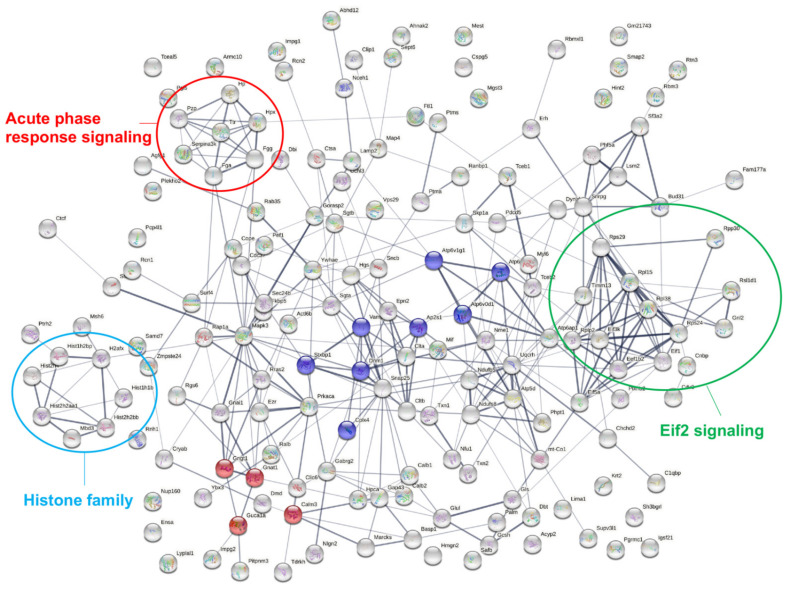
Interaction network analysis of all DEPs using STRING. The top two KEGG pathways are the “phototransduction” (mmu04744, molecules involved were labeled in red nodes) and the “synaptic vesicle cycle” (mmu04721, molecules involved were labeled in blue nodes). Several signaling pathways with molecules clustered together were labeled with large circles, including acute phase response signaling (red circle), eif2 signaling (green circle), and histone family (blue circle).

**Figure 5 ijms-23-06464-f005:**
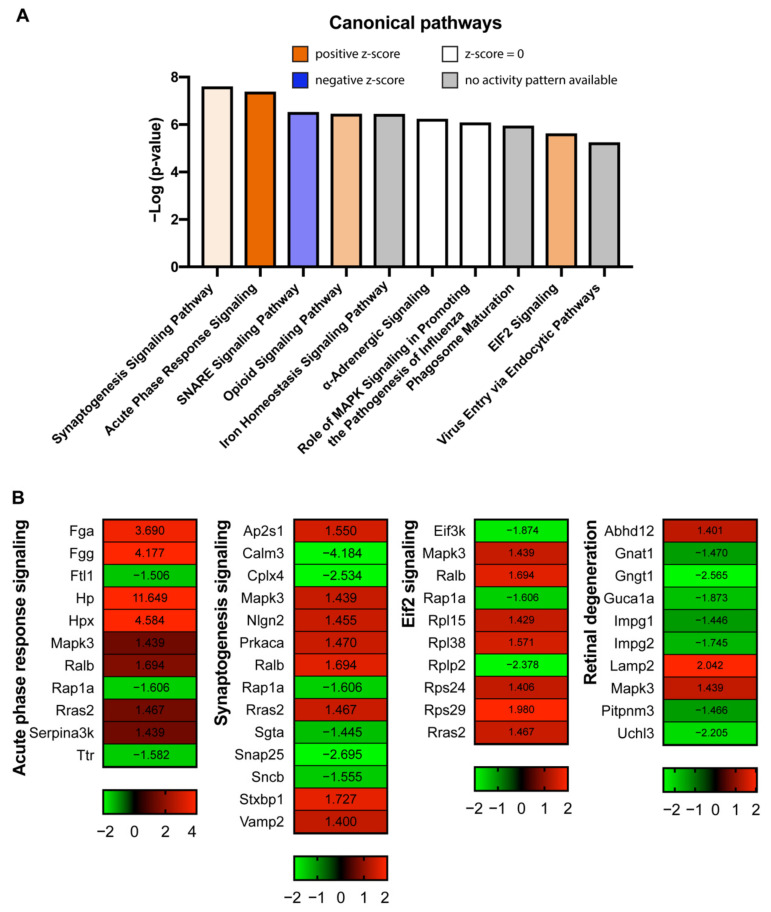
(**A**) The top 10 canonical pathways affected in the EIU model identified by IPA with −Log10 (*p*-value) and z-score indicated. Blue bars: negative z-score; orange bars: positive z-score; grey bars: no activity pattern available; white bars: z-score = 0. The darker coloring of the bars represents the more significant z-scores. (**B**) The heatmaps of canonical signaling pathways, including acute phase response signaling, synaptogenesis signaling, eif2 signaling, and retinal degeneration-related molecules. The number on each column represents the fold change of protein.

**Figure 6 ijms-23-06464-f006:**
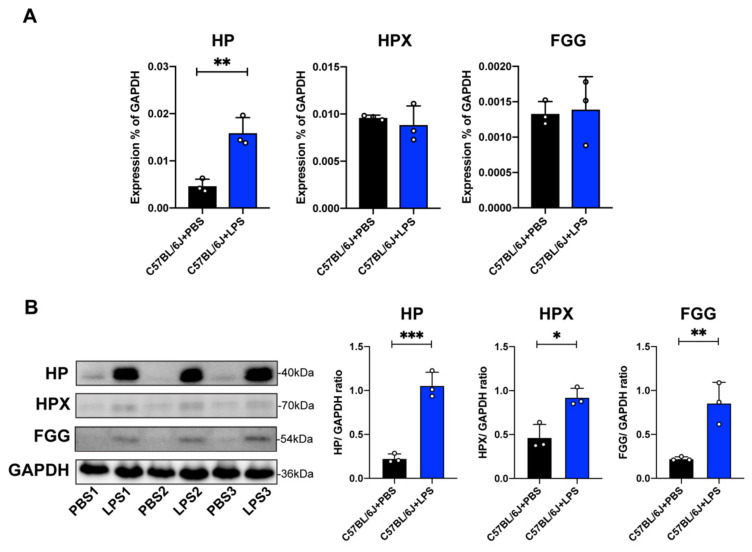
(**A**) The mRNA expression levels of haptoglobin (HP), hemopexin (HPX), and fibrinogen gamma chain (FGG) in the retina of C57BL/6J mice treated by LPS (n = 3) or PBS (n = 3). (**B**) The protein expression levels of retinal HP, HPX, and FGG in LPS-treated C57BL/6J mice (n = 3) or PBS-treated C57BL/6J control groups (n = 3) were measured by Western blot analysis. GAPDH levels were used as a loading control. Representative Western blot images were shown on the left, and quantification of protein levels (right panel) were normalized to GAPDH protein levels. Data are expressed as the mean ± SD. * *p* < 0.05, ** *p* < 0.01, *** *p* < 0.001.

**Figure 7 ijms-23-06464-f007:**
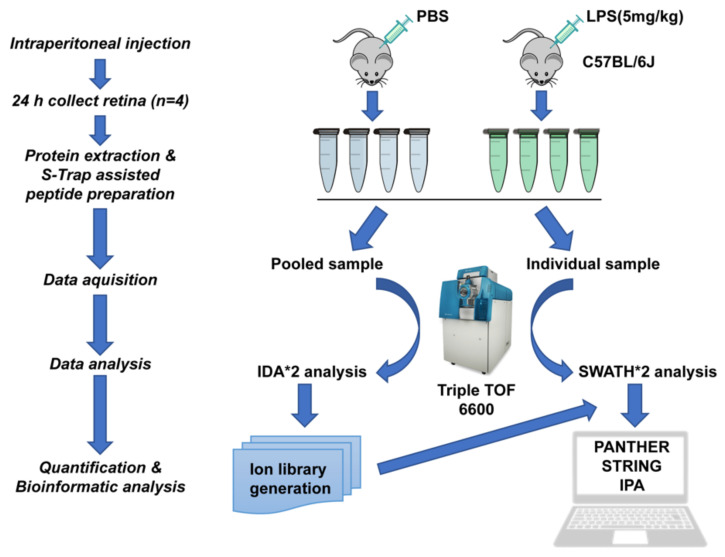
Schematic workflow of quantitative discovery proteomics in the retina from C57BL/6J mice treated with an intraperitoneal injection of LPS or PBS solution (n = 4 for each group) by SWATH-MS. The peptide was prepared based on S-Trap filter and trypsin digestion. The 2 μg of peptides from each retina sample was analyzed by a Triple-TOF 6600 MS. The acquired data were further analyzed with bioinformatic online tools including ProteinPilot, PANTHER, STRING, and IPA.

**Table 1 ijms-23-06464-t001:** Top 10 upregulated (red) or downregulated (green) DEPs.

*UniProt ID*	*Gene Name*	*Protein Name*	*Fold Change*	*p-Value*
B2RWH3	*Hist2h2aa1*	Histone H2A	13.450	0.003
Q3UBS3	*Hp*	Haptoglobin	11.649	0.000
Q91X72	*Hpx*	Hemopexin	4.584	0.000
P43274	*H14*	Histone H1.4	4.519	0.019
Q8CGP2	*Hist1h2bp*	Histone H2B type 1-P	4.344	0.010
Q3UER8	*Fgg*	Fibrinogen gamma chain	4.177	0.000
E9PV24	*Fga*	Fibrinogen alpha chain	3.690	0.000
Q64525	*Hist2h2bb*	Histone H2B type 2-B	3.307	0.013
Q9D0J8	*Ptms*	Parathymosin	3.082	0.007
Q1WWK3	*H1f5*	Hist1h1b protein	2.537	0.023
Q6W8Q3	*Pcp4l1*	Purkinje cell protein 4-like protein 1	−2.568	0.000
Q3TE95	*Rcn2*	Uncharacterized protein	−2.569	0.000
Q05186	*Rcn1*	Reticulocalbin-1	−2.642	0.000
P60879	*Snap25*	Synaptosomal-associated protein 25	−2.695	0.000
Q3U6E4	*Ptma*	Uncharacterized protein	−2.933	0.000
A0A087WPA0	*Lbhd1*	LBH domain-containing 1	−3.062	0.000
Q3TL58	*Skp1*	S-phase kinase-associated protein 1	−3.117	0.000
Q542N8	*Palm*	Uncharacterized protein	−3.286	0.000
P0DP28	*Calm3*	Calmodulin-3	−4.183	0.004
Q91XV3	*Basp1*	Brain acid soluble protein 1	−4.443	0.000

## Data Availability

The mass spectrometry proteomics data have been deposited to the ProteomeXchange Consortium via the PRIDE partner repository with the dataset identifier PXD033532.

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
