# Peer review of "Retinal Proteomic Analysis in a Mouse Model of Endotoxin-Induced Uveitis Using Data-Independent Acquisition-Based Mass Spectrometry"

_ijms, 2022, doi:10.3390/ijms23126464_

Round 1
Reviewer 1 Report
The paper is clearly written and describes a novel proteomic analysis of retinal proteins in a mouse model of uveitis using a data-independent acquisition-based mass spectrometry (SWATH-MS). The findings are of interest but there are aspects of presentation that require clarification.
In terms of methods, more information is required to ensure that the data analysis is clear in terms of how this was achieved.
Which software was used to plot the Volcano plot (Figure 2)- this should be stated in the methods.
Table S1 - the full list of proteins should include sequence coverage information
Fig. 6B - individual western blots. There are multiple bands/lane for HP, HPX, FGG - please edit the western blot to include an arrow indicating the immunoreactive band used for the quantitative analysis. There is no Figure legend so it is not possible to assess if a positive control (for the protein of interest) is included. This is essential to have confidence in the data shown in Fig. 6B in the manuscript. Please include a figure legend. Also direct the reader to the original western blot figures in the supplementary file.
For Fig. 6B - data is shown for 3 samples. 4 samples were used for SWATH-MS. Are the 3 samples used for western blot derived independently from the 4 for SWATH or these a subset of those used for proteomic analysis. Ideally, follow up analysis, using western blot is directed to both the samples used for the proteomic analysis and a separate set of independently derived samples.
Line 245 - the discrepancy between mRNA and protein data (Fig. 6A) is commented on but not interpreted. mRNA levels do not necessarily correlate to protein levels and a lack of correlation can be interesting.
The supplemental files refer to a video - line 471 - this is not referred to elsewhere in the text. Please delete if not relevant.
The proteomic data should be submitted to Proteome Exchange and be publicly available. Please do this and include the information for review.
Reviewer 2 Report
This paper reports the analysis of the retinal proteomic profiles of Endotoxin-induced uveitis and normal C57BL/6J mice using a data-independent acquisition-based mass spectrometry (SWATH-MS). The objectives of the study are well defined, and the introduction provides a state of art with appropriate references. Overall I consider that this article contains enough robust data to evidence the conclusions.
Minor
What was the rationale for the selection of the proteins for validation by qPCR and western blot analysis?
Why didn't you analyse the direct effects of the intraocular route for LPS application to compare the with the intraperitoneal route?
Reviewer 3 Report
Zhang et al. conducted experiments to study retinal proteome changes during in the pathogenesis of endotoxin-induced uveitis. They showed significantly regulated proteins and identified phototransduction and synaptic vesicle cycle as two of the major pathways regulated. The study provides some interesting findings, however, significantly lacks signaling studies. As potential mechanisms, they suggest some pathways, but these observations are descriptive, only based on gene expression levels or pathway analysis.
Specific comments
- The title of the MS needs to be revised as the entire proteomic study was from the posterior eye cups, which include retina, RPE and choroidal tissues.
- Mice should have been screened for Pde6Brd1 as well as rd8 as the presence of these mutations can produce significant disease phenotypes unrelated to the gene or genes of interest.
- Fig1 C. In addition to mRNA, authors should measure protein levels of IL-1β, IL-6, iNOs and TLR4. Authors should also include retinal H&E staining of the control and EIU.
- Fig. 6. Authors should show immunoblots of selected regulated proteins observed in each of these pathways. Provide complete blots as supplementary data to confirm the specificity of the antibodies used.
Round 2
Reviewer 1 Report
The manuscript is improved but there are still concerns
Specific responses
- Volcano plot - completed
- Table S1 - the sequence coverage has been added, please add a legend explaining the %Cov (50), %Cov (95). Since the number of matching peptides are shown with 95% confidence (only) - it's perhaps not essential to include %Cov (50) information.
- The western blot data remains to be addressed- A positive control sample is required/essential for assignment of the protein based on selection of the appropriate immunoreactive band.
- Please add this information on the source of the sample sets to the main text of the manuscript for evaluation by the reader.
- - 7. Completed
Additional comments
Figure 1, Page 4 - the legend indicates insertion of additional data in 2 panels - these are not in the Figure in the revised version.
D) ELISA analysis of IL-1β and IL-6 expression levels in the retina tissues of LPS- and PBS-treated groups (n=4).
(E) Western blotting analysis and densitometry of iNOS and TLR4 in the retina of LPS- and PBS-treated groups (n=3).
Table S3 - please provide information in the legend on the settings used in the STRINGS analysis - so that this is complete and comparable with the information in Table S4.
Reviewer 3 Report
The authors answered some of the concerns of the reviewer. However, there are still some issues to be resolved which would strengthen the MS.
Specific comments
Figure 1 E: Authors should show three complete western blots full images.
Fig 6 B: Authors showed only representative blots, should show all complete blots (n =3). Look like the antibodies are non-specific as there are bands throughout and other bands are much stronger than the protein the authors are looking for. To increase the reliability, the authors should repeat the Western blot analysis with suitable positive controls.
Round 3
Reviewer 1 Report
Thank you for the clear responses. The information on the western blot data is represents improvement but there is still a lack of clear labelling of the positive control. Complete western blot images, with the immunoreactive bands of interest indicated with an arrow in both the positive control and the samples tessted must be shown. For response 5- STRINGS - please add this to the methods section rather than supplemental information, section 4.9.
Reviewer 3 Report
Authors satisfactorily revised the MS
Author Response
Thank you for your review.